# Achieving destination sustainability: How tourist-to-tourist interaction quality affects tourist loyalty?

**Junli Gao[1], Fang Meng[2], Weifeng Guo[1]\*, Baomin Lin[3]**

**1** School of Tourism, Wuyi University, Nanping, China, **2** School of Hospitality and Tourism Management, College of Hospitality, Retail and Sport Management, University of South Carolina, Columbia, South Carolina, United States of America, **3** College of Tourism, Fujian Normal University, Fuzhou, Fujian, China

\* hnguoweifeng@126.com

**Data Availability Statement:** All relevant data files are available from the figshare repository (https://doi.org/10.6084/m9.figshare.23706654).

**Funding:** This study is funded by China National Social Science Foundation (19XGL010). The

## Abstract

Although tourist-to-tourist interaction (TTI) has been identified as an essential element in tourist experiences, the effect of TTI quality on tourist loyalty, and the mechanism underlying this effect is scarcely discussed in the literature. Based on the self-determination theory, this study aims to examine whether and how TTI quality influences tourist loyalty, representing a significant means to achieve destination sustainability. More specifically, this study tested a moderated mediation model in which basic psychological needs satisfaction mediated the relationship between TTI quality and tourist loyalty, while sociability moderated the link between TTI quality and basic psychological needs satisfaction. A survey research approach was used, and 746 complete, usable responses were collected in multiple cities in China. The results revealed that the direct impact of TTI quality on tourist loyalty is mediated by basic psychological needs satisfaction. Furthermore, sociability positively moderates the influence of TTI quality on tourist loyalty. This study extends the TTI literature by demonstrating the mechanism of basic psychological needs satisfaction and tourists' sociability in the relationship between TTI quality and tourist loyalty. Managerial suggestions are provided for industry practitioners to improve tourist relationship management and the sustainability of destinations.

## Introduction

Social interaction, the act or practice of mutual contact and influence [1] is shown to effectively improve interpersonal relations, produce positive individual emotions, and reduce sadness, fatigue, pain, and other negative psycho-social states [2]. Tourism provides a platform for individuals from different cultural backgrounds to participate in various forms of interpersonal interactions [3]. Three main types of tourist interaction have been identified: tourists and local communities, tourists and service providers, and tourists and other tourists [4]. Tourist-to-tourist interaction (TTI) is a core element that shapes tourists' evaluation of their travel experience and satisfaction [5–8]. More importantly, in the circumstance of fierce competition in

funders had no role in study design, data collection and analysis, decision to publish, or preparation of the manuscript. No author received a salary from the funder.

**Competing interests:** The authors have declared that no competing interests exist.

destinations [9, 10], TTI itself creates a tourist experience beyond the enjoyment of scenic spots or the types of destinations [8, 11, 12]. Therefore, it is critical for both academics and industry practitioners to examine TTI and its impact on tourist experience and behaviors.

Extant studies have mainly focused on exploring the outcome variables of TTI in different tourism contexts. For example, TTI is reported to be conducive to a better overall tourist experience and more satisfaction in a group tour [12, 13]. Research also suggested that backpackers' interactions with fellow travelers are important in forming internal identity, destination choice, and the overall travel experience [14, 15]. More importantly, prior literature revealed that TTI positively affects tourist satisfaction [6, 16, 17]; which is an essential factor in inducing tourist loyalty [18–20]. Accordingly, TTI quality may potentially trigger tourist loyalty toward destinations and eventually achieve destination sustainability. Sustainable development of a destination refers to a balanced development in the economic, social, cultural, and environmental dimensions of the destination. Tourist loyalty is crucial for maintaining tourist relations, reducing the costs associated with acquiring tourists [21], and ensuring stable tourist arrivals and revenue [22]. These factors are vital for the sustainable development of destinations in economic, social and cultural aspects [23]. Therefore, it is essential to expound the relationship between TTI quality and tourist loyalty for tourist relationship management and long-term sustainability of destinations.

Despite its importance, the extant literature has neglected the effect of TTI quality on tourist loyalty, and the mechanism underlying this effect is scarcely discussed. TTI is originated from customer–customer interaction (CCI), which is found to trigger customers' emotional attachment and subsequently influences loyalty [24, 25]. In the tourism context, individuals are away from their home environment and have the need to interact with new people during their travel [26]. The hedonic nature of travel and interactions are associated with psychological needs of making friends, expressing themselves, exchanging tourism information, and obtaining a sense of identity [12, 14, 15]. However, the relationship between TTI quality and tourists' basic psycho-logical needs satisfaction and tourist loyalty has been scarcely discussed [27–29], which is not conducive to promoting the sustainable development of destinations.

In response, this study employs the self-determination theory to explore the in-fluence of TTI quality on tourist loyalty from the perspective of tourists' basic psycho-logical needs satisfaction. Under the framework of self-determination theory [30–32], the present study examines the connection between three constructs: TTI quality, basic psychological needs satisfaction, and tourist loyalty. In this study, we highlight the importance of relationships based on the self-determination theory, and posit that high quality relationships provide bond among individuals (i.e., relatedness) and reinforce people's needs for autonomy and competence. In this study, TTI quality represents an antecedent variable that affects the satisfaction of tourists' basic psychological needs. Tourist loyalty is a behavioral reaction after tourists' satisfaction of their basic psychological needs is achieved. Considering the importance of interpersonal interaction [7, 12, 28], this study proposes that TTI quality promotes tourist loyalty through the impact of basic psychological needs satisfaction. In other words, the fulfilment of basic psychological needs plays a critical mediating role between TTI quality and tourist loyalty. In addition, as interpersonal interaction is closely related to individuals' personality traits of sociability, this study proposes that the process in which TTI quality influence basic psycho-logical needs satisfaction is moderated by sociability [33–35].

Based on the variables discussed above, this study constructs a conceptual model that includes TTI quality, sociability, basic psychological needs satisfaction, and tourist loyalty (Fig 1). Three research aim are to be achieved in this study: (1) to examine whether and how TTI quality influences tourist loyalty; (2) to examine the mediating effect of basic psychological needs satisfaction on the relationship between TTI quality and tourist loyalty; (3) to examine

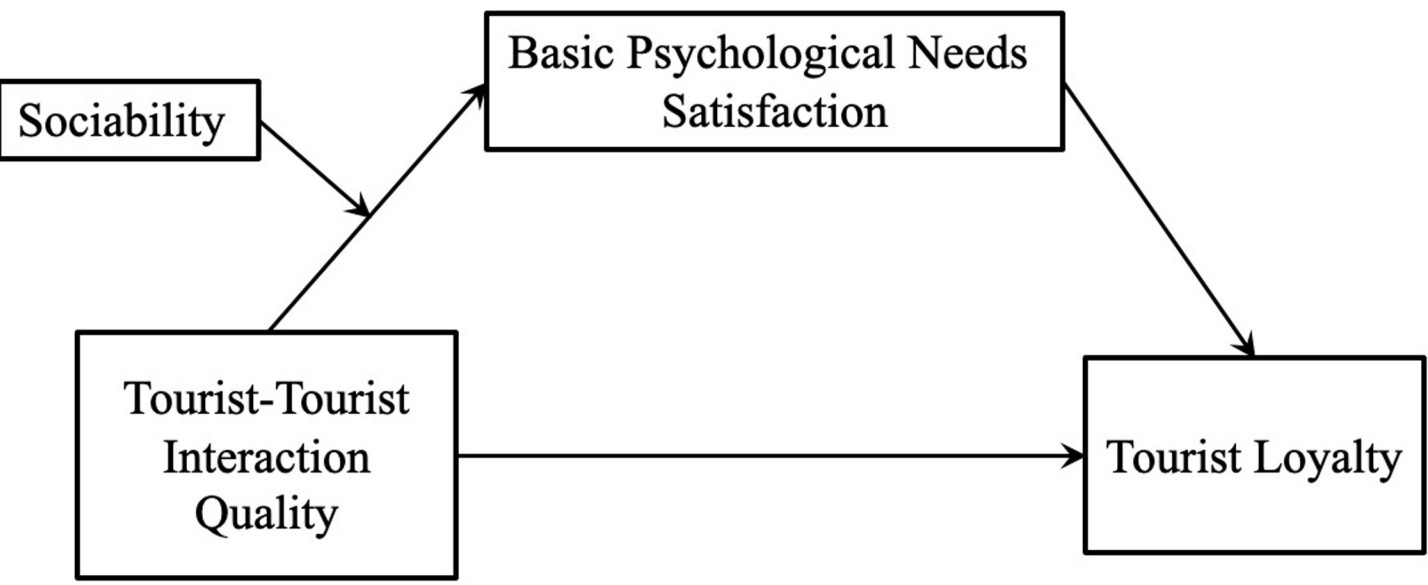

**Fig 1. Theoretical model.**

the moderating effect of sociability on the relationship between TTI quality on tourists' basic psychological needs satisfaction. This study contributes to the existing research in three ways. First, unlike previous literature on the antecedents of tourists' loyalty from the individual and destination level [27, 36–38], this study investigates the driving factor of tourists' loyalty from a perspective of tourist-tourist relationship. Second, this study enriches the TTI literature by clarifying the psychological mechanism of TTI quality stimulating tourist loyalty through basic psychological needs satisfaction. Third, the research model constructed in this study offers a analytical framework for future study to test others similar variables.

## Literature review and hypotheses development

**Tourist-to-tourist interaction quality.** TTI reflects the exchange of information, emotions, and feelings among tourists [11], such as providing assistance or advice, creating fun, embracing, and so on [7]. TTI is considered a multidimensional construct [7] but there has been no consensus on the measurement of TTI. For example, TTI was measured by related variable such as self-disclosure [12], or based on the valence of the interaction experience (positive or negative) [8, 39], tourist contact mode (verbal or non-verbal) [28], or quantity and quality [16, 40]. Comparatively, the quantity and quality of interaction have been widely applied in tourism settings [16, 41–43], indicating that this measurement has certain applicability and stability. More importantly, previous studies have shown that the effect of interaction quality on individual behavior or perception is often disturbed by interaction quantity [16, 44, 45]. Thus, the current study solely focuses on the effect of interaction quality but excludes interaction quantity. Specifically, TTI quality in this study refers to tourists' subjective perception of interactions that reflects their travel experiences related to other tourists [16].

**Tourist-to-tourist interaction quality and tourist loyalty.** Tourist loyalty, also often referred to "destination loyalty," is defined as the degree of tourist willingness to recommend and revisit specific destinations [46, 47], which is a means to eventually achieve destination sustainability. Previous research on the definition and measurement of tourist loyalty focused on three aspects: attitudinal, behavioral, and composite loyalty [48, 49]. Among them,

attitudinal loyalty refers to the degree of tourist willingness to recommend a destination [50]. Behavioral loyalty involves tourist consumption behavior at the destination, such as the frequency of repeated visitation [7, 49]. Composite loyalty, which is the integration of attitudinal and behavioral loyalty, refers to the dual intention to both revisit and recommend [48]. Given its crucial role in the success of destinations, tourist loyalty has received considerable attention [51, 52]. Research on the antecedents of tourist loyalty can be summarized as follows: (1) tourist-related factors, such as motivation [27, 29], involvement [53], and previous experience [23, 36, 54, 55]; (2) destination-related factors, such as destination image [37, 49], authenticity [56, 57], service quality [38, 58], e-service quality and e-recovery [59], residents' attitude and behavior [36, 37, 47–55]; and (3) post-travel factors, such as tourist experience and tourist satisfaction [22, 60, 61]. However, few scholars have investigated the relationship between TTI quality and tourist loyalty. In fact, TTI quality is an essential part of tourism experiences [7, 28] that is likely to promote the generation of tourist loyalty.

This study proposes that TTI quality is an important variable in the formation of tourist loyalty. First, TTI is an essential element of tourist experiences [7, 8]. The higher the quality of TTI, the more likely tourists have a satisfactory and pleasant experience [16, 62, 63], which further promotes tourists' recommendations and repeat visitation [22, 50]. Moreover, high TTI quality enhances tourist engagement by strengthening tourists' relations and identity [12], and tourists who have high involvement tend to achieve self-satisfaction from tourism activities and develop a stronger behavior intention [64, 65]. Thus, TTI quality is likely to positively influence tourist loyalty. Second, specific behaviors that generate TTI quality are of great value to tourists. For example, sharing information and experience [5, 7], providing effective countermeasures [66], and offering practical as-assistance help tourists better understand destinations and solve various problems during travel. Consequently, TTI quality may effectively promote tourist loyalty. Lastly, previous studies have indicated that tourists who experienced meaningful interactions are more inclined to show stronger behavioral intention [67]. Hence, this study proposes the following hypothesis:

H1: TTI quality positively affects tourist loyalty.

**The mediating effect of basic psychological needs satisfaction.**   Basic psychological needs satisfaction refers to the degree to which an individual ex-ternal environment satisfies one's basic psychological needs [68]. According to the self-determination theory, basic psychological needs satisfaction provides an analytical framework for understanding the generation mechanism of individual behavior [69, 70], which helps explain the mechanism of TTI quality affecting tourist loyalty.

This study proposes that TTI quality affects the satisfaction of tourists' basic psycho-logical needs. According to the self-determination theory, social environment factors (e.g., social support and interpersonal support) influence the satisfaction of the basic psycho-logical needs [71, 72]. Hence, in representing interpersonal interactions between tourists [28, 73], TTI quality contributes to tourists' basic psychological needs satisfaction. First, to some extent, tourist experience is interpersonal-oriented beyond destination-oriented [74]. TTI quality satisfies the motivation and needs of tourists through making friends and interacting with other tourists [18, 27]. Meanwhile, in a memorable tourism experience, tourists typically seek meaningful interactions such as exchanging their personal travel experience with other tourists, sharing tourism information, seeking help from others, and so on [7, 75]. These motivations and needs, which reflect tourists' free will, could be driven by their internal self-determination needs and satisfied by TTI quality. Second, achieving self-improvement and seeking a sense of identity are important determinants of TTI quality [76]. Studies have shown that strangers will respond to help-seekers' requests in specific situations to prevent the help-seeker from making a wrong decision [5]. Individuals who are knowledgeable about products are willing to help

(e.g., giving advice and im-parting knowledge) even before help-seekers ask for assistance, which in turn satisfies their own internal needs [77]. During the interaction, tourists enhance their reputation and gain social identity by displaying their talent and knowledge [76, 78]. There is no doubt that TTI quality supports and satisfies the individual's competency needs. Finally, hu-mans are essentially social [79] and have the need to pursue relationships, communication, and relatedness [80]. It is substantial for tourists to establish relatedness and social connection through frequent contact and shared experiences in travel to form self-identity [63, 81, 82]. Therefore, the following hypothesis is proposed:

H2: TTI quality has a positive effect on basic psychological needs satisfaction.

In addition, the satisfaction of tourists' basic psychological needs may also induce tourist loyalty. First, basic psychological needs satisfaction is often associated with individual mental state and vitality [83]. Tourists whose basic psychological needs are fulfilled have optimal psychological state [70] and more vigor to participate in tourism activities, and they are more likely to obtain an unforgettable tourism experience [65]. They also tend to develop recommendation behavior and future revisit behavior after the tour. Second, basic psychological needs satisfaction is significantly positively correlated with lei-sure satisfaction in tourism [84]. Tourists with high satisfaction are more willing to recommend and promote the destination to others [85, 86]. During travel, tourists achieve autonomy and competence, and consequently immerse themselves in the tourism environment more easily [87]. Additionally, for those who have satisfied the need for related-ness, friendship with other tourists undoubtedly adds value to the tourist experience [71]. Therefore, the following hypothesis is proposed:

H3: Tourists' basic psychological needs satisfaction has a positive impact on tourist loyalty.

Based on the above discussion, this study infers that the impact of TTI quality on tourist loyalty is mediated by the tourists' basic psychological needs satisfaction. From the perspective of self-determination theory, this study constructs an analysis path of "inter-action-needs satis-faction behavior." TTI quality promotes tourists' basic psychological needs satisfaction, and tourists show loyalty after the satisfaction of basic psychological needs. Hence, this study proposes the following hypothesis:

H4: Basic psychological needs satisfaction mediates the relationship between TTI quality and tourist loyalty.

**The moderating effect of sociability.** According to the self-determination theory, the process of environmental factors triggering basic psychological needs satisfaction is moderated by personality [33–35]. In other words, the extent to which TTI quality affects tourists' basic psychological needs satisfaction would rely heavily on individual characteristics. Sociability, which reflects individual differences in interpersonal relationship involvement [88], is proposed to influence tourists' interpersonal interaction. In this study, we investigate whether sociability moderates the link between TTI quality and tourists' basic psychological needs satisfaction.

Sociability refers to a tendency to affiliate with and prefer to be around others [89]. As a personality trait characteristic, sociability affects the personal evaluation of satisfaction and experience [90, 91]. Particularly, it plays a crucial role in moderating interpersonal interactions [88, 92]. Individuals with higher levels of sociability are similar to extroverts, not only in their tendency to seek friendships but also in their levels of engagement in various social interactions [88, 93]. Conversely, people who demonstrate low sociability are less responsive to external stimulation and prefer to focus on their inner worlds rather than socialize with others [88, 93].

In this study, we propose that sociability strengthens the link between TTI quality and tourists' basic psychological needs satisfaction. Highly sociable individuals tend to actively establish connections with others and enjoy social interaction [94, 95]. Moreover, they are prone to have a satisfying experience and receive positive feedback (e.g., affirmation, social integration)

in the process of interaction [28, 96]. Accordingly, tourists with high sociability are inclined to actively interact with others [94], which would improve TTI quality and better meet individuals' basic psychological needs. Thus, TTI quality evokes basic psychological needs satisfaction of tourists with high sociability, whereas individuals with low sociability prefer to be alone and avoid social interaction [95]. It means that their interaction with other tourists is transient and superficial and further weakens the positive effect of TTI quality on basic psychological needs satisfaction. Hence, we propose the following hypothesis:

H5: Sociability moderates the relationship between TTI quality and tourists' basic psychological needs satisfaction, such that the effect is stronger for tourists with high sociability.

We propose that tourists' basic psychological needs satisfaction mediates the relationship between TTI quality and tourist loyalty, and sociability moderates the influence of TTI quality on tourists' basic psychological needs satisfaction. According to the self-determination theory, basic psychological needs satisfaction is affected by individual characteristics in transmitting external environment-driven behavior [33]. Therefore, sociability is likely to affect the mediating effect of tourists' basic psychological needs satisfaction, which may involve a moderated mediating phenomenon [97]. Thus, we speculate that sociability also moderates the mediation of tourists' basic psychological needs satisfaction between TTI quality and tourist loyalty. Consequently, we propose that the indirect effect of the TTI quality on tourist loyalty through their basic psychological needs satisfaction should be stronger among tourists with high sociability. Based on the above analysis, we propose the following hypothesis:

H6: Sociability moderates the indirect effect of TTI quality on tourist loyalty mediated by basic psychological needs satisfaction such that the indirect effect is stronger for tourists with high sociability.

## Methods

### Data collection and sample

This study proposal was reviewed and approved by the Ethics Committee for Human Research Subject from the first authors' academic institution in Fujian, China. All the data collection procedures involving human subjects were conducted in accordance with the Ethics Committee's guidelines of the institution.

Participants were drawn from tourists who visited different destinations located in Fujian, China. To ensure representativeness of the samples, we took the following measures. Data were collected from tourists who had interacted with other tourists, particularly those participating in group tours. Specifically, through the authors' contacts in various travel agencies, we successfully obtained the cooperation of 12 professional tour guides who provide guided group tours in Fujian. Tourists who participated in their group tours were approached at the end of the trip regarding this study survey. They were informed that the questionnaire was to understand their tourism experiences, and that their participation was entirely voluntary. They were ensured that all their answers would be completely anonymous and confidential. Verbal consent was obtained from each participant. Only after the participant verbally agreed to fill in the survey and confirmed that they are above 18 years old, they were given the questionnaire. Upon completing the survey, each participant was given a local token as the incentive and appreciation for their time and input. Based on the procedure mentioned above, we randomly investigated tourists' interaction experience with other tourists according to the schedule of 12 tour guides from July 15th to December 23rd, 2022. Data indicated that participants have visited the different types of destinations in Fujian, such as urban tourism (e.g., Xia Men, Fu Zhou), mountain tourism (e.g., Nan Ping), coastal tourism (e.g., Zhang Zhou, Pin Tan), and Red tourism (e.g., Long Yan, San Ming).

We distributed 860 questionnaires in total during the data collection period and obtained 746 complete and usable responses after eliminating 114 invalid, incomplete responses, indicating a response rate of 86.74%. As shown in Table 1, of the 746 respondents, 55.8% were female, 36.5% were between the ages of 30 and 39, and 30% were between the ages of 40 and 49. Concerning the education, 75.3% of the respondents had received higher education. Regarding the income, 32.7% of the respondents had a monthly income of between ¥4,001 and ¥6,000, and 28.2% earned between ¥6,001 and ¥8,000 monthly.

## Measures

As this study was conducted in China, we followed Brislin's (1970) back-translation procedure to translate all the items into Chinese [98]. To guarantee the quality of the questionnaire, we invited two English tourism experts to revise the items to ensure the consistency of semantics and content. The variables in this study were scored on a 5-point Likert scale (from 1 = "disagree completely" to 5 = "agree completely"). The specific methods of measurement were as follows.

**Tourist-to-tourist (TTI) interaction quality.** We assessed TTI quality with Huang & Hsu's five-item scale (2010) [16]. Tourists reported their perception of interactions between tourists. Sample items included "I experienced an interesting interaction with other tourists during this trip" and "I experienced an equal interaction with other tourists during this trip." The scale's reliability was 0.88.

**Basic psychological needs satisfaction.** Basic psychological needs satisfaction was measured using Jiseon's (2019) scale that comprises three items: perceived autonomy [99], perceived competence, and perceived relatedness. Each of the three subscales contained three items. Tourists reported the extent to which they felt basic psychological needs satisfaction. Sample items included "This interaction makes me feel free to do things my way," "This interaction makes me feel competent," and "This interaction makes me feel a lot closer with others." The scale's reliability was 0.93.

**Sociability.** We adopted Cheek & Buss's (1981) five-item scale to measure sociability [89]. Sample items included "I like to be with people" and "I prefer working with others rather than alone." The scale's reliability was 0.88.

**Tourist loyalty.** We assessed tourist loyalty with Akhoondnejad's (2016) three-item scale [22]. Sample items included, "If I have a chance, I will revisit this destination," "I will recommend this destination to other people," and "I have a willingness to pay more in this destination." The scale's reliability was 0.87.

**Table 1. Sample profile (N = 746).**

| Demographic | Total number | Percentage | Demographic | Total number | Percentage |
|---|---|---|---|---|---|
| **Gender** | | | **Education** | | |
| Male | 330 | 44.2% | Junior high or below | 184 | 24.7% |
| Female | 416 | 55.8% | College | 244 | 32.7% |
| | | | Undergraduate | 270 | 36.2% |
| | | | Master or above | 48 | 6.4% |
| **Age (years)** | | | **Monthly income(RMB)** | | |
| <20 | 14 | 1.9% | <2000 | 58 | 7.8% |
| 21–29 | 136 | 18.2% | 2001–4000 | 176 | 23.6% |
| 30–39 | 272 | 36.5% | 4001–6000 | 244 | 32.7% |
| 40–49 | 224 | 30% | 6001–8000 | 210 | 28.2% |
| >50 | 100 | 13.4% | >6000 | 58 | 7.8% |

**Control variables.** Gender, age, education, monthly income, and other demographic characteristics were considered to impact tourist loyalty [29, 54]. Therefore, to avoid the influence of these irrelevant variables on the logical relationship between the variables in this study, we used gender, age, education, and monthly income as control variables.

## Analytic strategy

We employed SPSS PROCESS and Mplus 7.4 to verify and test the moderated mediation model constructed in this study. First, we used Mplus 7.4 for confirmatory factor analysis. Through analyzing the values of composite reliability and average variance extracted [100, 101], we examined the reliability and validity of the model constructed in this study. Meanwhile, we checked the influence of common method variance drawing support from preliminary statistical verification [102]. Furthermore, with the aid of SPSS PROCESS macro models 7 [103], we investigated the direct effect in the proposed model and examined whether sociability moderated the indirect link between TTI quality and tourist loyalty mediated by basic psychological needs satisfaction. Additionally, we performed a simple slope analysis to illustrate the moderating effect of sociability [104].

## Results

### Confirmatory factor analysis

To investigate the discriminant validity of TTI quality, basic psychological needs satisfaction, sociability, and tourist loyalty, we utilized Mplus 7.4 to conduct confirmatory factor analysis on the data obtained. As shown in Table 2, the four-factor model including TTI quality, sociability, basic psychological needs satisfaction, and tourist loyalty [$\chi^2(203)$ = 748.52, comparative fit index [(CFI) = 0.95, Tucker–Lewis index (TLI) = 0.94, root mean square error of approximation (RMSEA) = 0.06, and standardized root mean square residual (SRMR) = 0.04] demonstrated higher goodness of fit than any other alternative model including the single-factor model [$\chi^2(209)$ = 4394.23, CFI = 0.59, TLI = 0.54, RMSEA = 0.16, and SRMR = 0.15]. As shown in Table 3, factor loadings of all variables ranged from 0.71 to 0.86, which indicated good internal consistency. The composite reliability value for each construct was higher than 0.7, indicating that the four-factor model had satisfactory reliability [101]. All constructs' average variance extracted (AVE) values were greater than 0.5, indicating adequate convergent validity [100]. Additionally, the square roots of AVE scores for each factor were greater than

**Table 2. Results of confirmatory factor analysis.**

| Model | Factor structure | $\chi^2$ | df | $\chi^2$/df | CFI | TLI | RMSEA | SRMA |
|---|---|---|---|---|---|---|---|---|
| Five-factor model | TTI, SO, BPNS, TL, CMV | 653.60 | 181 | 3.61 | 0.95 | 0.94 | 0.06 | 0.03 |
| Four-factor model | TTI, SO, BPNS, TL | 748.52 | 203 | 3.69 | 0.95 | 0.94 | 0.06 | 0.04 |
| Three-factor model 1 | TTI+TL, BPNS, SO | 1498.09 | 201 | 7.45 | 0.87 | 0.85 | 0.09 | 0.06 |
| Three-factor model 2 | TTI+SO, BPNS, TL | 2532.77 | 206 | 12.30 | 0.77 | 0.74 | 0.12 | 0.12 |
| Three-factor model 3 | TTI, BPNS, SO+TL | 1984.40 | 206 | 9.63 | 0.83 | 0.80 | 0.11 | 0.14 |
| Two-factor model 1 | BPNS, TTI+SO+TL | 3259.85 | 208 | 15.67 | 0.70 | 0.67 | 0.14 | 0.13 |
| Two-factor model 2 | TTI+BPNS, SO+TL | 2979.16 | 208 | 14.32 | 0.73 | 0.70 | 0.13 | 0.16 |
| Single-factor model | TTI+BPNS+SO+TL | 4394.23 | 209 | 21.03 | 0.59 | 0.54 | 0.16 | 0.15 |

Note: N = 746. TTI = tourist-to-tourist interaction quality; SO = sociability; BPNS = basic psychological needs satisfaction; TL = tourist loyalty; CMV = common method variance; CFI = comparative fit index; TLI = Tucker-Lewis index; RMSEA = root mean square error of approximation; SRMR = standardized root mean square residual. "+" means combining into one factor.

**Table 3. Results of the measurement model.**

| Construct | Mean | SD | Skewness | Kurtosis | Loading | Alpha | AVE |
|---|---|---|---|---|---|---|---|
| TTI quality | | | | | | 0.88 | 0.60 |
| I have a harmonious relationship with other tourists during this trip. | 3.71 | 1.12 | -0.86 | 0.18 | 0.81 | | |
| I have experienced an interesting interaction with other tourists during this trip. | 3.73 | 1.03 | -0.70 | 0.02 | 0.76 | | |
| I have experienced an equal interaction with other tourists during this trip. | 3.78 | 1.01 | -0.82 | 0.26 | 0.73 | | |
| I have a cooperative relationship with other tourists during this trip. | 3.74 | 1.08 | -0.75 | 0.04 | 0.75 | | |
| I have experienced an intense interaction with other tourists during this trip. | 3.72 | 1.09 | -0.75 | -0.12 | 0.82 | | |
| Basic psychological needs satisfaction | | | | | | 0.93 | 0.61 |
| I feel free to do things my way. | 3.78 | 1.11 | -0.93 | 0.25 | 0.79 | | |
| I free to be who I am. | 3.80 | 1.10 | -0.97 | 0.32 | 0.75 | | |
| I feel free during this trip. | 3.84 | 1.08 | -0.98 | 0.36 | 0.80 | | |
| I feel competent. | 3.82 | 0.96 | -0.53 | -0.29 | 0.73 | | |
| I feel capable. | 3.79 | 1.09 | -1.07 | 0.68 | 0.84 | | |
| I feel effective | 3.85 | 1.08 | -0.95 | 0.45 | 0.78 | | |
| I feel a lot of closeness with other tourists. | 3.86 | 1.08 | -0.94 | 0.32 | 0.73 | | |
| I feel a lot of closeness with other tourists. | 3.77 | 1.12 | -0.88 | 0.17 | 0.79 | | |
| I feel very close to other tourists. | 3.80 | 1.14 | -0.93 | 0.17 | 0.81 | | |
| Sociability | | | | | | 0.88 | 0.60 |
| I like to be with people. | 3.69 | 0.94 | -0.51 | 0.16 | 0.73 | | |
| I welcome the opportunity to mix socially with people. | 3.54 | 0.98 | -0.50 | 0.10 | 0.71 | | |
| I prefer working with others rather than alone. | 3.41 | 1.04 | -0.20 | -0.54 | 0.72 | | |
| I find people more stimulating than anything else. | 3.54 | 0.98 | -0.27 | -0.31 | 0.84 | | |
| I'd be unhappy if I were prevented from making many social contacts. | 3.57 | 1.02 | -0.54 | 0.02 | 0.84 | | |
| Tourist Loyalty | | | | | | 0.87 | 0.68 |
| I will say positive things about this destination to other people. | 3.93 | 1.14 | -1.01 | 0.20 | 0.86 | | |
| I will keep visiting this destination if possible in the future. | 3.93 | 1.10 | -1.09 | 0.49 | 0.82 | | |
| I will recommend this destination to my relatives and friends. | 3.91 | 1.04 | -1.03 | 0.68 | 0.80 | | |

Note. N = 746. TTI = Tourist-to-tourist interaction; SD = standard deviation; AVE = average variance extracted.

their corresponding inter-construct correlation (see Table 4), suggesting adequate discriminant validity [100]. The above results indicated that the four latent variables conducted in this study had sufficient discriminant validity and distinctly represented four different constructs.

Given that the measurements of TTI quality, basic psychological needs satisfaction, sociability, and tourist loyalty were completed by tourists using a questionnaire, there may be a risk of common method variance when investigating the relationship among variables. Thus, we examined the influence of common method variance (CMV) by controlling for unmeasured latent method factors [102, 105]. We introduced a latent variable, CMV, into the four-factor model and allowed all items of the four latent variables (TTI quality, basic psychological needs satisfaction, sociability, and tourist loyalty) to load on it. Compared with the four-factor model, all the fitting index values in the five-factor model were not significantly improved (less than 0.02), indicating that CMV slightly affected subsequent analyses.

## Descriptive statistics

Table 4 presents the results of descriptive statistics, including standard deviations, means, and correlations of all variables in this research. The results showed that TTI quality correlated positively with basic psychological needs satisfaction (γ = 0.59, P<0.01), sociability (γ = 0.16,

Table 4. Means, standard deviations and correlations of all variables.

| Variables | 1 | 2 | 3 | 4 | 5 | 6 | 7 | 8 |
|---|---|---|---|---|---|---|---|---|
| 1.Gender | | | | | | | | |
| 2.Age | 0.07* | | | | | | | |
| 3.Education | 0.01 | -0.47** | | | | | | |
| 4.Income | -0.10** | 0.15** | 0.19** | | | | | |
| 5.TTI quality | -0.02 | 0.05 | -0.08* | 0.08* | 0.77a | | | |
| 6.Basic psychological needs satisfaction | 0.01 | -0.01 | -0.07 | 0.06 | 0.59** | 0.78a | | |
| 7. Sociability | -0.01 | 0.03 | -0.05 | 0.14** | 0.16** | -0.03 | 0.77a | |
| 8. Tourist loyalty | 0.08* | -0.03 | 0.05 | 0.08* | 0.47** | 0.44** | 0.22** | 0.82a |
| *Mean* | 1.56 | 3.35 | 2.24 | 3.05 | 3.74 | 3.81 | 3.55 | 3.92 |
| *Standard deviance* | 0.50 | 0.99 | 0.90 | 1.07 | 0.88 | 0.87 | 0.81 | 0.98 |

Note: N = 746. a = square root of AVE; *P<0.05(2-tailed); **P<0.01(2-tailed)

P<0.01), and tourist loyalty (γ = 0.47, P<0.01). Basic psychological needs satisfaction was positively related to tourist loyalty (γ = 0.44, P<0.01). In addition, sociability was positively related to tourist loyalty (γ = 0.22, P<0.01) and was insignificantly correlated with basic psychological needs satisfaction (γ = -0.03, ns). The insignificant relation-ship between sociability and basic psychological needs satisfaction did not affect the sub-sequent regression analysis.

## Hypotheses testing

Table 5 displays the results of the moderated mediation analysis. Taking the control variables (gender, age, education, monthly income) and mediator variable (basic psychological needs satisfaction) into account, the main effect of TTI quality on tourist loyalty was statistically significant (β = 0.37, 95% confidence interval [CI] = [0.28, 0.45]). Thus, H1 was supported.

As shown in Table 5, TTI quality was positively related to basic psychological needs satisfaction (β = 0.61, P < 0.01), and the 95% CI did not include zero (0.56, 0.67), thus supporting H2. Meanwhile, basic psychological needs satisfaction significantly impacted tourist loyalty (β = 0.28, 95% CI = [0.19, 0.36]), as displayed in Table 5, which supported H3. In addition, we introduce bootstrapping (5,000) to test the indirect effect of TTI quality on tourist loyalty. The result of bootstrapping (5,000) revealed that the indirect effect was significant (β = 0.17, P < 0.01), and the 95% CI excluded zero (0.11, 0.24), thus supporting H4.

Table 5 shows that the interaction term between TTI quality and sociability had a significant and positive impact on basic psychological needs satisfaction (β = 0.14, P < 0.01), with the 95% CI excluding zero (0.08, 0.20). According to West & Aiken (1991), we used a simple slope analysis by plotting the pattern of moderating effects at one standard deviation above and below the mean. As displayed in Fig 2, the influence of TTI quality on basic psychological needs satisfaction among tourists with high sociability is stronger than those with low sociability. Therefore, H5 was supported.

Table 5 demonstrates that the mediated effect of basic psychological needs satisfaction increased for tourists with high sociability (β = 0.20, 95% CI = [0.13, 0.28]). However, for tourists with low sociability, the influence of TTI quality on tourist loyalty via basic psychological needs satisfaction was weaker (β = 0.14, 95% CI = [0.08, 0.21]). More importantly, the moderated mediation at high and low levels of sociability showed a significant difference (β = 0.04, P < 0.05), and the 95% CIs did not include zero (0.02, 0.06), which supported H6.

**Table 5. Results of regression analyses.**

| Variables | Basic Psychological Needs satisfaction | | | Tourist loyalty | | |
|---|---|---|---|---|---|---|
| | β | SE | 95% CI | β | SE | 95% CI |
| **Constant** | **3.99**** | **0.17** | **(3.67, 4.32)** | **2.36** | **0.27** | **(1.84, 2.88)** |
| Control variable | | | | | | |
| Gender | 0.07 | 0.05 | (-0.03, 0.17) | 0.16** | 0.06 | (0.04, 0.29) |
| Age | -0.08** | 0.03 | (-0.14, -0.02) | -0.01 | 0.04 | (-0.08, 0.06) |
| Education | -0.08* | 0.03 | (-0.15, -0.02) | 0.09* | 0.04 | (0.01, 0.17) |
| Income | 0.05* | 0.03 | (0.01, 0.10) | 0.03 | 0.03 | (-0.03, 0.09) |
| Independent variable | | | | | | |
| TTI quality | 0.61** | 0.03 | (0.56, 0.67) | 0.37** | 0.04 | (0.28, 0.45) |
| Mediator variable | | | | | | |
| Basic psychological needs satisfaction | | | | 0.28** | 0.04 | (0.19, 0.36) |
| Moderator variable | | | | | | |
| Sociability | -0.12** | 0.03 | (-0.18, -0.05) | | | |
| Moderating Effect | | | | | | |
| TTI quality × sociability | 0.14** | 0.03 | (0.08, 0.20) | | | |
| Mediating effects | | | | | | |
| Basic psychological needs satisfaction (Low- sociability) | | | | 0.14 | 0.03 | (0.08, 0.21) |
| Basic psychological needs satisfaction (Mean- sociability) | | | | 0.17 | 0.03 | (0.11, 0.24) |
| Basic psychological needs satisfaction (High- sociability) | | | | 0.20 | 0.04 | (0.13, 0.28) |
| Index of moderated mediation | | | | 0.04 | 0.01 | (0.02, 0.06) |
| R2 | 0.39 | | | 0.28 | | |
| F | 33.68** | | | 23.44** | | |

Note: N = 746. TTI = tourist-to-tourist interaction; SE = standard error. 95%CI = Bias-corrected confidence interval. *P<0.05; **P<0.01.

# Conclusions

## Discussion of the results

According to the findings of this study, one can surmise that TTI quality is a crucial predictor of tourist loyalty. Previous research on the antecedents of tourist loyalty mainly focused on individual and destination levels [23, 37, 38], neglecting the impact of TTI quality. Specially, previous studies have found that consumers' e-exchange, e-service quality, and e-recovery are determining factor of consumer loyalty [53, 59]. This enlightens us on whether offline individual's engagement could evoke their loyalty. This study revealed that TTI quality indeed is a key influence factor promoting tourist loyalty, which enriches the literature on tourist loyalty. In addition, prior studies on the effect of TTI on tourist behavior mainly concentrated on interaction mode [28], interaction valence [39], and interaction content [7], but they rarely discussed the impact of TTI from the perspective of interaction quality. Based on the existing literature on the quality of tourist-tourist interaction [17], this study further investigated the transmission mechanism of TTI quality triggering tourist loyalty, thereby extending the research on TTI.

This study also revealed that basic psychological needs satisfaction is a critical path connecting TTI quality and tourist loyalty. The existing research on TTI mostly inherits the paradigm and framework of CCI [106]. Tracking back the research on the relationship between CCI and consumer loyalty, it seemed that there are two equally practical path: cognitive evaluation and emotional response [5, 24]. However, both cognitive and emotional perspective primarily focused on individual response under interactive situation, ignoring the influence of interactive experience on basic psychological needs satisfaction. The result provides support for

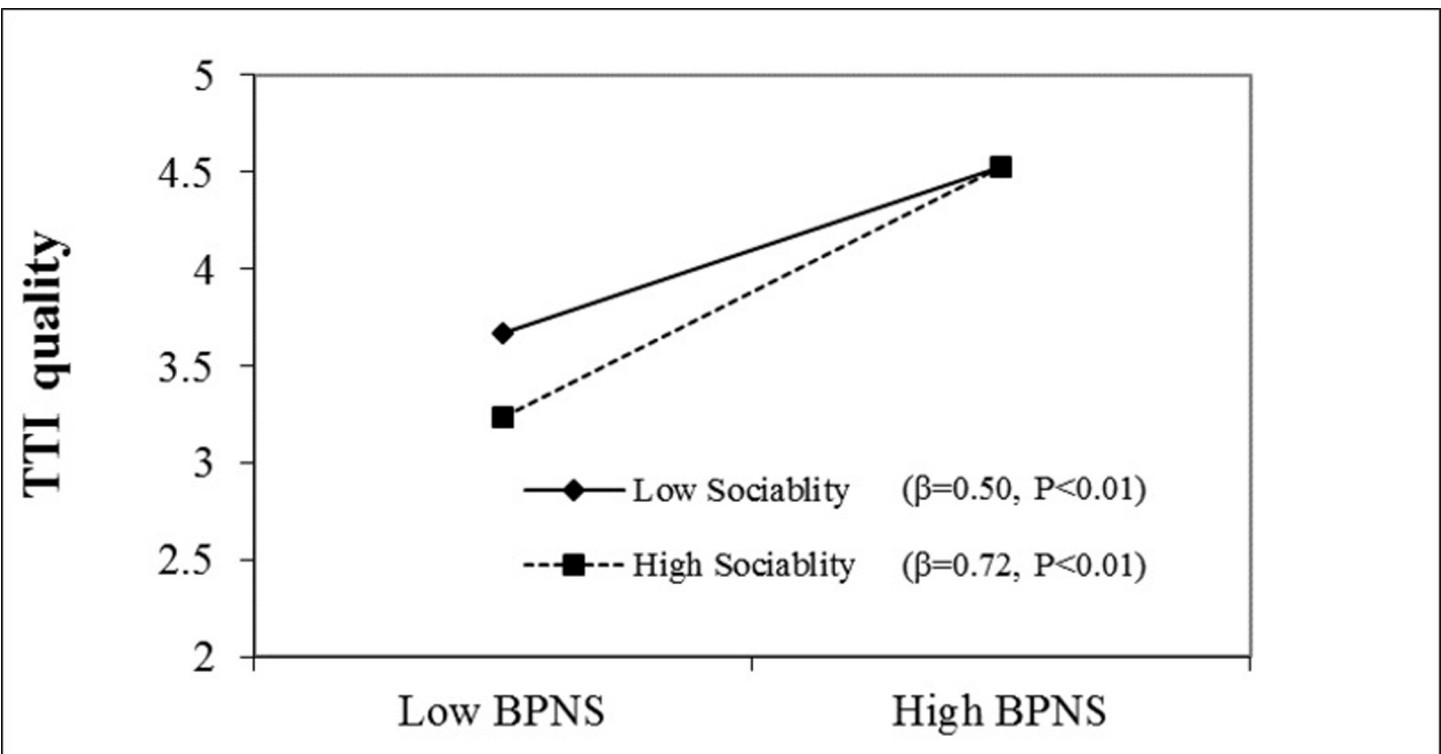

**Fig 2. Moderating effects of sociability on the relationship between TTI (tourist-to-tourist interaction) quality and BPNS (basic psychological needs satisfaction).**

statement that TTI arouses tourist behavior [28] and expands the knowledge about how such a relation works from the perspective of self-determination theory.

Self-determination theory posits that conditions supporting the individual's experience of autonomy, competence, and relatedness would foster individuals' motivations and engagement for activities, including enhanced performance and persistence [30–32]. Particularly, based on the self-determination theory, individuals' interactions among each other are desirable and essential, as the relationships provide satisfaction of the need for relatedness. Furthermore, the high-quality relationships not only satisfy the relatedness need but also fulfill the autonomy need and competence need. Specifically, findings of this study are consistent with the self-determination theory and strongly support the following conclusions: (1) TTI quality promotes tourists' basic psychological needs satisfaction; (2) tourist loyalty is a behavioral response to basic psychological needs satisfaction; and (3) a psychological needs satisfaction perspective provides a reliable explanation for understanding the influence process of TTI quality. These findings are in line with the statement reported in tourism that basic psychological needs satisfaction can be induced by individual experience and promote behavioral intention [107]. In addition, previous studies on the mediating effect of basic psychological needs satisfaction mostly focused on the fields of psychology, management, and pedagogy [33, 108, 109], paying less attention to tourists' basic psychological needs satisfaction. This study extends prior findings by empirically analyzing the transmission mechanism of tourists' basic psychological needs satisfaction from the perspective of tourist-tourist interaction.

This study also examines the role of sociability in the process of tourists' interactive experience on their basic psychological needs satisfaction. Although studies have pointed that sociability is a moderating variable worthy of attention in tourism [92], researchers have not examined its impact on tourist-tourist relationship. Moreover, previous research primarily

concentrated on the main effect of sociability, paying less attention to its moderating effect. For instance, Li et al. (2021) revealed that social personality trait can significantly stimulate consumers' purchase behavior [91]. Different from existing literature, the current study confirmed that sociability also acts as moderating role in the link between TTI quality and basic psychological needs satisfaction.

## Theoretical implications

The theoretical implications of this study are as follows. First, this study provides insights into the process of how tourists' interactions trigger their post-travel behavior. Previous research has acknowledged that TTI is an integral part of the tourist experience and affects tourists' on-site behavior [28], but it largely ignored the effect of TTI quality on tourists' post-travel behavior. Responding to the call for investigation of TTI using empirical research [12], this study explored the relationship between TTI quality and tourist loyalty. Findings of this study confirmed that TTI quality stimulates tourist loyalty, which suggests that TTI has an impact on tourists' on-site and post-travel behavior. In addition, previous literature on tourist loyalty mainly focused on its antecedent at the individual and destination level [27, 36, 53, 59]. This study analyzes the impact of TTI quality on tourist loyalty from the tourist-tourist relationship perspective, thus enriching the literature on tourist loyalty, which is a means to the sustainability of tourism destinations.

Second, this study advances the TTI literature by revealing the mediating mechanism underlying the relationship between TTI quality and tourist loyalty. Previous research discussed the effect of TTI from different dimensions [12, 28], but little research concerns the direct impact of TTI quality on tourist loyalty. Meanwhile, previous literature on the mechanism of CCI stimulating consumer loyalty mostly concentrated on individual response in the interactive situation [5, 24, 25], neglecting the satisfaction effect of interactive behavior on individuals' basic psychological needs. Hence, through introducing basic psychological needs satisfaction as the mediator, our study clarifies the psychological mechanism of TTI quality in arousing tourist loyalty. Furthermore, considering the lack of research on basic psychological needs satisfaction in tourism studies [107, 110], the investigation of its mediating mechanism in TTI literature widens the scope of basic psycho-logical needs satisfaction. In addition, given the insufficient research on the moderating role of tourist sociability, the current study addressed this gap by investigating the moderating effect of tourist sociability on the relationship between TTI quality and tourist basic psychological needs satisfaction, which complements the existing research.

Finally, we constructed a moderated mediation model based on self-determination theory, offering a theoretical framework for exploring and testing similar variable in the field of TTI. This study's research model comprises four constructs (TTI quality, individual-al basic psychological needs satisfaction, behavioral reaction after basic psychological needs satisfaction, and individual characteristic). The theoretical framework of this study can also be used to explore other similar variables in future studies. For example, TTI may not always be beneficial. Negative TTI reflects tourists' negative evaluation of experience related to other tourists [41, 111] and may be regarded as a negative environmental variable. Likewise, given that basic psychological needs can also be frustrated by external environment [112], basic psychological needs frustration represent a strong and direct threat to the needs [113], which can be further verified in future studies.

## Managerial implications

This study presents practical implications for the management of group tours. First, previous studies indicated that TTI plays an essential role in shaping the tourism experience [8, 12, 28]. Thus, necessary intervention measures that enhance TTI quality should be taken, such as designing high-quality group interaction activities, providing interactive games that require collaboration among tourists, and encouraging self-introductions among tourists to deepen mutual understanding. More importantly, compared with interaction quantity, tourism service providers should pay more attention to the formation of high-quality TTI rather than interaction frequency. To this end, they can focus on some key indicators of interaction quality (e.g., harmonious; friendly; equal; cooperative; and intense interactions) to roughly evaluate TTI quality.

Second, this study finds that basic psychological needs satisfaction plays a mediating role between TTI quality and tourist loyalty. Hence, taking intervention measures that can increase tourists' basic psychological needs satisfaction is beneficial to service providers. For example, service providers can improve tourist basic psychological needs satisfaction by guiding high-quality TTI. Additionally, tourists' basic psychological needs may be challenged in an unfamiliar tourism destination [28]. Thus, tourism practitioners should take action to increase tourists' autonomy, competence, and relatedness needs satisfaction by paying attention to the physical and psychological state of tourists. Specifically, they should respect and satisfy tourists' autonomy and competence needs without interfering with the regular schedule, and design tourism activities involving more inter-actions to satisfy tourists' needs for relatedness and strengthen their bond to the destination.

In addition, this study revealed that sociability positively moderates the impact of TTI quality on tourist loyalty. As tourists who demonstrate low sociability prefer to engage in solitary activities rather than seeking interactions, tourism enterprises and destination managers should emphasize activities for those with low sociability. To avoid the resistance of low sociability tourists, destination managers and tourism practitioners should ascertain and respect their willingness (or unwillingness) to be around others when arranging interactive activities. They need to ensure non-force participation and make alternative arrangements for tourists who wish not to engage in such activities. Moreover, Ruan (2020) indicated that groups with similar sociability contribute to satisfaction [92]. Therefore, matching tourists' sociability when designing interactive games or tourism activities would be effective to improve tourist satisfaction.

## Limitations and future research

Although this study enriches the existing knowledge about TTI, there are some limitations to be addressed in future research. First, we assessed TTI through the quality of interaction, which has been applied in prior studies [15, 41, 114]. However, TTI is considered a multidimensional construct, with different dimensions leading to different outcomes [7, 41]. Future research should investigate the influence mechanism of other types of TTI on tourist loyalty and explore the differences in various types of TTI on tourist loyalty. Second, through exploring the link between TTI quality and basic psychological needs satisfaction, the present study finds that TTI quality is an antecedent of basic psychological needs satisfaction, but it ignores the motivation that drives TTI; that is, tourists may seek TTI to fulfil basic psychological needs. Thus, it would be meaningful to clarify the relationship among basic psychological needs, TTI quality, and basic psychological needs satisfaction in the future. Third, influenced by collective values in China [115], the conclusions are in line with self-determination theory and our hypotheses. However, previous studies implied that individual behavior would be

different due to the national culture [116], which may be reflected in TTI. Hence, it is a new direction worth exploring that uses a cross-cultural analysis to investigate the effect of TTI quality to get more generalizable results. Fourth, the COVID-19 pandemic may have altered tourists' interactions and how the TTI quality is evaluated [117, 118]. Therefore, future studies should assess tourists' interaction changes and the impact on destination management in the post-pandemic era, which would further influences the long-term sustainable development of tourism destinations.

## Supporting information

**S1 Checklist. Human participants research checklist.**
(DOCX)

## Author Contributions

**Conceptualization:** Junli Gao.

**Formal analysis:** Baomin Lin.

**Funding acquisition:** Weifeng Guo.

**Investigation:** Weifeng Guo.

**Methodology:** Junli Gao.

**Project administration:** Junli Gao.

**Software:** Baomin Lin.

**Writing – original draft:** Junli Gao.

**Writing – review & editing:** Fang Meng.

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
