## [Decision Letter · Decision Letter 0]

18 Oct 2023

PONE-D-23-22635Achieving Destination Sustainability: How Tourist-To-Tourist Interaction Quality Affects Tourist Loyalty?PLOS ONE

Dear Dr. Guo,

Thank you for submitting your manuscript to PLOS ONE. After careful consideration, we feel that it has merit but does not fully meet PLOS ONE’s publication criteria as it currently stands. Therefore, we invite you to submit a revised version of the manuscript that addresses the points raised during the review process. Reviewer #1: Thank you for conducting such interesting research. Although it has several advantages, there are also some issues that you should resolve before further processing. The most important issue in your work is its sources which are not enough and should add more related sources to support your sentences and ideas as I mentioned bellow.

1. In the introduction, when you are speaking about the tourism, I expected to start your introduction with a wider view into the tourism importance by the following sources to support your idea:

Nourbehesht, H., & Mohammad Shafiee, M. (2020). Strategic foresight of vulnerabilities of the tourism industry with focus on economic sanctions. Defensive Future Studies, 5(18), 113-140.

Hajiabadi, J. R., Shafiee, M. M., & Kazemi, A. (2021). The impact of tourism destination brand experience on value co-creation by focusing on the mediating role of destination brand love. Journal of Brand Management, 7(4), 89-118.2. Please also elaborate more on your research gap as well as the research contribution in the last section of the introduction.

3. I expect to see your research aim and/or research question in the last section of the introduction as well.

4. Moreover, the current review of literature is not strong and should be improved by using the above mentioned sources I have proposed. I also recommend the following sources to improve the heading “Tourist-to-tourist interaction quality and tourist loyalty”:

Mohammad Shafiee, M., & Ahghar Bazargan, N. (2019). The impact of E-exchange development on customers’ E-loyalty and repurchase intention. Journal of Business Administration Research, 10(20), 71-90.

Shafiee, M. M., & Bazargan, N. A. (2018). Behavioral customer loyalty in online shopping: The role of e-service quality and e-recovery. Journal of theoretical and applied electronic commerce research, 13(1), 26-38.

5. Again in the above heading, where you wrote:

“Research on the antecedents of tourist loyalty can be summarized as follows: (1) tourist-related factors, such as motivation [24,26], involvement, and previous experience [48-50]; (2) destination-related factors, such as destination image [44,51], authenticity [52-53], service quality [54-55], residents’ attitude and behavior [42-51]; and (3) post-travel factors, such as tourist experience and tourist satisfaction [20,56-57].

I expect to see more supporting source for each of these statements. For example, you can add the following sources to strengthen your statements:

6. In the method section, you should also add some valid sources for the mentioned procedure you have used in the paper. In the current format there is no sources to support your method procedure.

7. Moreover in this section I would like to see a table containing your questionnaire items and their sources as well as the credibility indices if the items such as Alpha etc.

8. In the method section you should be more clear by mentioning the period and method of data gathering, validity etc.

9. The data analysis section is well done and complete. However you should be more concise in some section by removing the redundant sentences and explanations.

10. All tables and figures should have complete format of the abbreviations and terms.

11. Please compare your results in the discussion section with previous literature, especially the sources I have suggested to you.

12. Your conclusion section should be stronger by providing more practical implications for managers and practitioners as well as future researchers.

13. Final sources have some minor issues and should be according to the journal format.

Good luck with your revised paper.

Reviewer #2: Please improve the discussion part

Research methodology should also be revised and more details may be added.

In introduction part update the references Social media destination information features and destination loyalty: does perceived coolness and memorable tourism experiences matter?

You can use above research

We look forward to receiving your revised manuscript.

Kind regards,

Wilmer Carvache-Franco, PhD.

Academic Editor

PLOS ONE

Journal Requirements:

"Funding:This study is funded by China National Social Science Foundation (19XGL010)."

"Conflicts of Interest: The authors received no specific funding for this work"

Reviewers' comments:

Reviewer's Responses to Questions

**Comments to the Author**

1. Is the manuscript technically sound, and do the data support the conclusions?

Reviewer #1: Yes

Reviewer #2: Yes

2. Has the statistical analysis been performed appropriately and rigorously? 

Reviewer #1: Yes

Reviewer #2: Yes

3. Have the authors made all data underlying the findings in their manuscript fully available?

Reviewer #1: Yes

Reviewer #2: Yes

4. Is the manuscript presented in an intelligible fashion and written in standard English?

Reviewer #1: Yes

Reviewer #2: Yes

5. Review Comments to the Author

Reviewer #1: Thank you for conducting such interesting research. Although it has several advantages, there are also some issues that you should resolve before further processing. The most important issue in your work is its sources which are not enough and should add more related sources to support your sentences and ideas as I mentioned bellow.

1. In the introduction, when you are speaking about the tourism, I expected to start your introduction with a wider view into the tourism importance by the following sources to support your idea:

Nourbehesht, H., & Mohammad Shafiee, M. (2020). Strategic foresight of vulnerabilities of the tourism industry with focus on economic sanctions. Defensive Future Studies, 5(18), 113-140.

Hajiabadi, J. R., Shafiee, M. M., & Kazemi, A. (2021). The impact of tourism destination brand experience on value co-creation by focusing on the mediating role of destination brand love. Journal of Brand Management, 7(4), 89-118.

Mohammad Shafiee, M., Tabaeeian, R. A., & Tavakoli, H. (2018). The effect of memorable brand experience of tourism destination on destination brand love with the mediating role of consumer-brand identification: Study of tourists in Isfahan. Journal of Tourism and Development, 7(3), 127-141.

Bakhshayesh, R., Mohammad Shafiee, M., & Kazemi, A. (2023). Destination quality, destination brand identification and behavioral intentions: A mixed method approach. Journal of Tourism and Development, 12(1), 25-42.

Shafiee, M. M., Tabaeeian, R. A., & Khoshfetrat, A. (2020). Tourist engagement and citizenship behavior: The mediating role of relationship quality in the hotel industry. Tourism and Hospitality Research, 20(4), 481-492.

Sanayei, A., Mohammad Shafiee, M., Shams, H., & Golchin, H. (2012). Effects of ICT on Marketing Mix in Electronic Tourism: Shaping Marketing Strategies in E-Tourism Enterprises, 6th International conference on ECDC, Shiraz, IEEE.

Shafiee, M. M., Rahimzadeh, S., & Haghighizade, R. (2016). The effect of implementing SEO techniques and websites design methods on e-tourism development: A study of travel agencies e-tourism websites. In 2016 10th International Conference on e-Commerce in Developing Countries: with focus on e-Tourism (ECDC) (pp. 1-8). IEEE.

Shafiee, M. M., & Najafabadi, S. I. (2016). The interaction of technological progress and tourism industry development in the developing countries: The case of Iran's tourism industry. In 2016 10th International Conference on e-Commerce in Developing Countries: with focus on e-Tourism (ECDC) (pp. 1-5). IEEE.

Please use all or most of these to improve your introduction and to review the research gap. In the current format, your introduction is not adequate.

2. Please also elaborate more on your research gap as well as the research contribution in the last section of the introduction.

3. I expect to see your research aim and/or research question in the last section of the introduction as well.

4. Moreover, the current review of literature is not strong and should be improved by using the above mentioned sources I have proposed. I also recommend the following sources to improve the heading “Tourist-to-tourist interaction quality and tourist loyalty”:

Mohammad Shafiee, M., & Ahghar Bazargan, N. (2019). The impact of E-exchange development on customers’ E-loyalty and repurchase intention. Journal of Business Administration Research, 10(20), 71-90.

Shafiee, M. M., & Bazargan, N. A. (2018). Behavioral customer loyalty in online shopping: The role of e-service quality and e-recovery. Journal of theoretical and applied electronic commerce research, 13(1), 26-38.

5. Again in the above heading, where you wrote:

“Research on the antecedents of tourist loyalty can be summarized as follows: (1) tourist-related factors, such as motivation [24,26], involvement, and previous experience [48-50]; (2) destination-related factors, such as destination image [44,51], authenticity [52-53], service quality [54-55], residents’ attitude and behavior [42-51]; and (3) post-travel factors, such as tourist experience and tourist satisfaction [20,56-57].

I expect to see more supporting source for each of these statements. For example, you can add the following sources to strengthen your statements:

For experience:

Mohammad Shafiee, M., Tabaeeian, R. A., & Tavakoli, H. (2018). The effect of memorable brand experience of tourism destination on destination brand love with the mediating role of consumer-brand identification: Study of tourists in Isfahan. Journal of Tourism and Development, 7(3), 127-141.

Hajiabadi, J. R., Shafiee, M. M., & Kazemi, A. (2021). The impact of tourism destination brand experience on value co-creation by focusing on the mediating role of destination brand love. Journal of Brand Management, 7(4), 89-118.

Bakhshayesh, R. S., Shafiee, M. M., & Kazemi, A. (2021). The impact of destination brand awareness and experience on destination brand identification. Journal of Brand Management, 8(4), 103-130.

Mohammad Shafiee, M., Foroudi, P., & Tabaeeian, R. A. (2021). Memorable experience, tourist-destination identification and destination love. International Journal of Tourism Cities, 7(3), 799-817.

For image:

Mohammad Shafiee, M., Tabaeeian, R., & Tavakoli, H. (2020). The effect of destination image on tourist satisfaction, intention to revisit and WOM: An empirical research in Foursquare social media. In 10th International Conference on e-Commerce in Developing Countries: With Focus on e-Tourism (ECDC). (pp. 1-8). IEEE.

For service quality:

Mohammad Shafiee, M., Tavakoli, H., & Tabaeeian, R. A. (2018). The effect of market orientation, service innovation and service quality on brand preference and willingness to pay higher prices: study of Rail transport companie’s passengers. Quarterly Journal of Brand Management, 5(1), 169-204.

Yavari, Z., Mohammad Shafiee, M., & Ghauor, F. (2018). Evaluating the service quality of selected specialized medical clinics in Shiraz city, Iran, using servqual model. Health Information Management, 14(6), 236-242.

Shafiee, M. M., & Bazargan, N. A. (2018). Behavioral customer loyalty in online shopping: The role of e-service quality and e-recovery. Journal of theoretical and applied electronic commerce research, 13(1), 26-38.

Tabaeeian, R. A., Mohammad Shafiee, M., & Ansari, A. (2023). Developing a scale for gamified e-service quality in the e-retailing industry. International Journal of Retail & Distribution Management, 51(4), 444-464.

For residents’ related factors:

Shafiee, M. M., Shafiee, M. M., Shams, H., Yahai, M. R., & Golchin, H. (2013). ICT capacities in creating sustainable urban tourism and its effects on resident quality of life. In 7th International Conference on e-Commerce in Developing Countries: with focus on e-Security (pp. 1-11). IEEE.

6. In the method section, you should also add some valid sources for the mentioned procedure you have used in the paper. In the current format there is no sources to support your method procedure.

7. Moreover in this section I would like to see a table containing your questionnaire items and their sources as well as the credibility indices if the items such as Alpha etc.

8. In the method section you should be more clear by mentioning the period and method of data gathering, validity etc.

9. The data analysis section is well done and complete. However you should be more concise in some section by removing the redundant sentences and explanations.

10. All tables and figures should have complete format of the abbreviations and terms.

11. Please compare your results in the discussion section with previous literature, especially the sources I have suggested to you.

12. Your conclusion section should be stronger by providing more practical implications for managers and practitioners as well as future researchers.

13. Final sources have some minor issues and should be according to the journal format.

Good luck with your revised paper.

Reviewer #2: Please improve the discussion part

Research methodology should also be revised and more details may be added.

In introduction part update the references Social media destination information features and destination loyalty: does perceived coolness and memorable tourism experiences matter?

You can use above research

6. PLOS authors have the option to publish the peer review history of their article (what does this mean?). If published, this will include your full peer review and any attached files.

Reviewer #1: No

Reviewer #2: No

---

## [Author Response · Author response to Decision Letter 0]

25 Nov 2023

Dear Editor，

We are truly grateful to you for giving us a chance to resubmit our paper. And we also would like to thank yours and other reviewers’ critical comments and thoughtful suggestions, which would help us to improve the quality of the paper. Based on these comments and suggestions, we have made careful modifications on the original manuscript. Here we submit a new version of our manuscript with the title “Achieving Destination Sustainability: How Tourist-To-Tourist Interaction Quality Affects Tourist Loyalty?”. We mark all the changes in red in the revised manuscript. And we hope the new manuscript will meet the standard of PLOS ONE. You will find our point-by-point responses to the reviewers’ comments/ questions beneath.

Sincerely yours,

The authors

Feedback to comments of reviewer 1

Thank you for conducting such interesting research. Although it has several advantages, there are also some issues that you should resolve before further processing. The most important issue in your work is its sources which are not enough and should add more related sources to support your sentences and ideas as I mentioned bellow. We so appreciate for your comments and thanks very much for your time and efforts in the process of our manuscript reviewing. Based on your comments, we will do our best to modify our paper to improve our manuscript quality. Below are our point-by-point responses (in blue). Thanks again for your kind suggestions. 

Q1: In the introduction, when you are speaking about the tourism, I expected to start your introduction with a wider view into the tourism importance by the following sources to support your idea:

Nourbehesht, H., & Mohammad Shafiee, M. (2020). Strategic foresight of vulnerabilities of the tourism industry with focus on economic sanctions. Defensive Future Studies, 5(18), 113-140.

Hajiabadi, J. R., Shafiee, M. M., & Kazemi, A. (2021). The impact of tourism destination brand experience on value co-creation by focusing on the mediating role of destination brand love. Journal of Brand Management, 7(4), 89-118. 

R1: Thanks very much for your very constructive suggestions. 

The two articles enlighten us on the relationship between tourism importance and TTI. We added this into the first paragraph of the “introduction” (See line 38-41). Detailed as following:

More importantly, in the circumstance of fierce competition in the destination (Hajiabadi et al., 2021; Nourbehesht & Mohammad Shafiee, 2020), TTI itself creates a tourist experience beyond the enjoyment of scenic spots at destinations [8-10]. 

See line 38-41

Q2: Please also elaborate more on your research gap as well as the research contribution in the last section of the introduction. 

R2: We so appreciate for your advice. Following your suggestion, we further illustrate the research gap and research contribution in the last section of the “introduction”: (See line 96-102)

This study contributes to the existing research in three ways. 

First, unlike previous literature on the antecedents of tourists’ loyalty from the individual and destination level [24,49,51,55], this study investigates the driving factor of tourists’ loyalty from a perspective of tourist-tourist relationship.

Second, this study enriches the TTI literature by clarifying the psychological mechanism of TTI quality stimulating tourist loyalty through basic psychological needs satisfaction. 

Third, the research model constructed in this study offer a analytical framework for future study to test others similar variables 

Q3: I expect to see your research aim and/or research question in the last section of the introduction as well. 

R3: Thanks very much for your very constructive suggestions. To clearly state the research aim of this study, we summarized three research question in the last of the “introduction” (See line 89-96). As follows:

Based on the variables discussed above, this study constructs a conceptual model that includes TTI quality, sociability, basic psychological needs satisfaction, and tourist loyalty (see Figure 1). Three research aim are to be achieved in this study: (1) to examine whether and how TTI quality influences tourist loyalty; (2) to examine the mediating effect of basic psychological needs satisfaction on the relationship between TTI quality and tourist loyalty; (3) to examine the moderating effect of sociability on the relationship between TTI quality on tourists’ basic psychological needs satisfaction.

Q4: Moreover, the current review of literature is not strong and should be improved by using the above mentioned sources I have proposed. I also recommend the following sources to improve the heading “Tourist-to-tourist interaction quality and tourist loyalty”:

Mohammad Shafiee, M., & Ahghar Bazargan, N. (2019). The impact of E-exchange development on customers’ E-loyalty and repurchase intention. Journal of Business Administration Research, 10(20), 71-90.

Shafiee, M. M., & Bazargan, N. A. (2018). Behavioral customer loyalty in online shopping: The role of e-service quality and e-recovery. Journal of theoretical and applied electronic commerce research, 13(1), 26-38.

Q5. Again in the above heading, where you wrote:

“Research on the antecedents of tourist loyalty can be summarized as follows: (1) tourist-related factors, such as motivation [24,26], involvement, and previous experience [48-50]; (2) destination-related factors, such as destination image [44,51], authenticity [52-53], service quality [54-55], residents’ attitude and behavior [42-51]; and (3) post-travel factors, such as tourist experience and tourist satisfaction [20,56-57].

I expect to see more supporting source for each of these statements. For example, you can add the following sources to strengthen your statements: 

R4R5: We so appreciate for your advice. 

The literature that you have mentioned have improved our understanding of the research on customer loyalty. Based on this, we added these two articles in the current review of tourist loyalty(See line 132-138). Specifically, “Research on the antecedents of tourist loyalty can be summarized as follows: (1) tourist-related factors, such as motivation [24,26], involvement (Mohammad Shafiee & Ahghar Bazargan, 2019), and previous experience [Jamshidi et al., 2023; 48-50]; (2) destination-related factors, such as destination image [44,51], authenticity [52-53], service quality [54-55], e-service quality and e-recovery (Shafiee & Bazargan, 2018), residents’ attitude and behavior [42-51]; and (3) post-travel factors, such as tourist experience and tourist satisfaction [20,56-57].”

Moreover, we also cited it to strengthen statements on elsewhere of this study. (e.g., see line 487-489 ) 

Q6: In the method section, you should also add some valid sources for the mentioned procedure you have used in the paper. In the current format there is no sources to support your method procedure. 

R6: We so appreciate for your comments. 

According to your suggestion, we have added some valid sources for the mentioned procedure in this study. For example, in the section of “measure”, we adopted Brislin's (1980) back-translation procedure to translate all the items into Chinese (see line 292-293). In the section of “Analytic strategy” and “result”, we have complemented some valid sources for analytic method have used in this study, such as, composite reliability (see line 325-326, 346-347) and average variance extracted (see line 325-326, 347-349), common method variance (see line 327-329), SPSS PROCESS macro models (see line 329-330), a simple slope analysis (see line 330-333, 395-397). 

Q7: Moreover in this section I would like to see a table containing your questionnaire items and their sources as well as the credibility indices if the items such as Alpha etc. 

R7: We so appreciate for your advice. 

Based on your constructive advice, we have filled out the Table 2 again. As Table 2 presented, it showed the means, standard deviations, skewness, kurtosis, loading, Alpha and AVE among the variables. (see table 2)

Q8. In the method section you should be more clear by mentioning the period and method of data gathering, validity etc. 

R8: Thanks very much for your constructive suggestions. In the section of “Data collection and sample”, we further illustrate the period and method of data gathering in detail (see line 262-280). First, data were collected from tourists who had interacted with other tourists, particularly tourists participating in group tours. Second, utilizing the advantage of school’s platform, we successfully recruited 12 tour guides that provide tour guide services for tourists to Fujian in a long-term, and surveyed their tourists when the journey is finished. Based on the procedure mention above, we randomly investigated tourists’ interaction experience with other tourists according to the schedule of 12 tour guides from July 15th to December 23rd 2022. Data indicated that participants have visited the different types of destinations in Fujian, such as urban tourism (e.g., Xia Men, Fu Zhou), Mountain tourism (e.g., Nan Ping), Coastal Tourism (e.g., Zhang Zhou, Pin Tan), and Red Tourism (e.g., Long Yan, San Ming), which avoid the interference of destination types on research results to some extent. 

Q9. The data analysis section is well done and complete. However you should be more concise in some section by removing the redundant sentences and explanations. 

R9: Thanks very much for your very constructive suggestions. Following your suggestion, we have improved the writing through removing the redundant sentences and explanations in the section of “The data analysis section”, especially in “Hypotheses testing” (see line 380-460). We have deleted the content of Hypotheses, and direct report the data results supporting Hypotheses. 

Q10. All tables and figures should have complete format of the abbreviations and terms.

R10: Thanks very much for your advice. According to your advice, we have rechecked the format of the abbreviations and terms in tables and figures. 

Q11. Please compare your results in the discussion section with previous literature, especially the sources I have suggested to you. 

R11: We so appreciate for your advice. In the section of “discussion”, we have compared your results with previous literature, especially the sources you have suggested to us (see line 425-430). Specially, previous studies have found that consumers’ E-exchange and e-service quality and e-recovery are determining factor of consumer loyalty (Mohammad Shafiee & Ahghar Bazargan, 2019; Shafiee & Bazargan, 2018). This enlighten that whether offline individual’s engagement could evoke their loyalty. This study revealed that TTI quality indeed is a key influence factor promoting tourist loyalty, which enriches the tourist loyalty literature. 

Q12. Your conclusion section should be stronger by providing more practical implications for managers and practitioners as well as future researchers. 

R12: Thanks for your professional query. Based on your suggestion, we have enhanced the statement in the section of “Theoretical implications” and “Managerial implications”. First, in the section of “Theoretical implications” (see line 513-520), we argued that the theoretical framework of this study can also be used to explore other similar variables in future studies. For example, TTI may not always be beneficial. Negative TTI reflects tourists’ negative evaluation of experience related to other tourists [36,105] and may be regarded as a negative environmental variable. Likewise, given that basic psychological needs can also be frustrated by external environment [106], basic psychological needs frustration represent a strong and direct threat to the needs [107], which can be further verified in future studies. Second, in the section of “Managerial implications”, we provided more practical implications for managers (see line 525-527, 529-531, 535-536, 538-543, 547-554). 

Q13. Final sources have some minor issues and should be according to the journal format.

R13: We so appreciate for your advice. We have rechecked and revised references of this study according to the journal format of PLOS ONE.

With the information mentioned above, we wish your questions were solved well. Thanks again for your thoughtful advice.

Feedback to comments of reviewer 2

 We so appreciate your comments and thank you very much for your time and efforts in the process of our manuscript reviewing. Based on your comments, we will do our best to modify our paper to improve our manuscript quality. Below are our point-by-point responses (in blue). Thanks again for your kind suggestions. 

Q1: Please improve the discussion part. 

R1: Thanks very much for your very constructive suggestions. According to your comment, we have improved the discussion part of this study in three aspects. First, in the section of “discussion”, we have compared the results with previous literature, such as the research on tourist loyalty (see line 425-430), TTI (see line 439-444), basic psychological needs satisfaction (see line 459-466), and sociability (see line 468-476). line 425-430

Q2: Research methodology should also be revised and more details may be added. 

R2: We so appreciate for your advice. Based on your suggestion, we have improve the research methodology in two ways. First, we have added some valid sources for the mentioned procedure in this study. For example, in the section of “measure”, we adopted Brislin's (1980) back-translation procedure to translate all the items into Chinese (see line 292-293). In the section of “Analytic strategy” and “result”, we have complemented some valid sources for analytic method have used in this study, such as, composite reliability (see line 325-326, 346-347) and average variance extracted (see line 325-326, 347-349), common method variance (see line 327-329), SPSS PROCESS macro models (see line 329-330), a simple slope analysis (see line 330-333, 395-397). Second, In the section of “Data collection and sample”, we provided more detail about data gathering (see line 262-280). For example, data were collected from tourists who had interacted with other tourists, particularly tourists participating in group tours; utilizing the advantage of school’s platform, we successfully recruited 12 tour guides that provide tour guide services for tourists to Fujian in a long-term, and surveyed their tourists when the journey is finished. Based on the procedure mention above, we randomly investigated tourists’ interaction experience with other tourists according to the schedule of 12 tour guides from July 15th to December 23rd 2022. Data indicated that participants have visited the different types of destinations in Fujian, such as urban tourism (e.g., Xia Men, Fu Zhou), Mountain tourism (e.g., Nan Ping), Coastal Tourism (e.g., Zhang Zhou, Pin Tan), and Red Tourism (e.g., Long Yan, San Ming), which avoid the interference of destination types on research results to some extent. line 292-293

Q3: In introduction part update the references Social media destination information features and destination loyalty: does perceived coolness and memorable tourism experiences matter?

You can use above research 

R3: Thanks very much for your advice. The article your mention deepens our understanding of the research on destination loyalty. Therefore, we not only added its into the introduction part (see line 54-57), but update the literature review on tourist loyalty in the section of “Tourist-to-tourist interaction quality and tourist loyalty” (see line 132-138) and “discussion part” (see line 423-425) line 54-57

With the information mentioned above, we wish your questions were solved well. Thanks again for your thoughtful advice.

---

## [Decision Letter · Decision Letter 1]

29 May 2024

PONE-D-23-22635R1Achieving Destination Sustainability: How Tourist-To-Tourist Interaction Quality Affects Tourist Loyalty?PLOS ONE

Dear Dr. Guo,

Thank you for submitting your manuscript to PLOS ONE. After careful consideration, we feel that it has merit but does not fully meet PLOS ONE’s publication criteria as it currently stands. Therefore, we invite you to submit a revised version of the manuscript that addresses the points raised during the review process.

We look forward to receiving your revised manuscript.

Kind regards,

Vanessa Carels

Staff Editor

PLOS ONE

Reviewers' comments:

Reviewer's Responses to Questions

**Comments to the Author**

1. If the authors have adequately addressed your comments raised in a previous round of review and you feel that this manuscript is now acceptable for publication, you may indicate that here to bypass the “Comments to the Author” section, enter your conflict of interest statement in the “Confidential to Editor” section, and submit your "Accept" recommendation.

Reviewer #1: All comments have been addressed

Reviewer #2: All comments have been addressed

Reviewer #3: All comments have been addressed

Reviewer #4: (No Response)

2. Is the manuscript technically sound, and do the data support the conclusions?

Reviewer #1: Yes

Reviewer #2: Yes

Reviewer #3: Yes

Reviewer #4: Yes

3. Has the statistical analysis been performed appropriately and rigorously? 

Reviewer #1: Yes

Reviewer #2: Yes

Reviewer #3: Yes

Reviewer #4: Yes

4. Have the authors made all data underlying the findings in their manuscript fully available?

Reviewer #1: Yes

Reviewer #2: Yes

Reviewer #3: Yes

Reviewer #4: Yes

5. Is the manuscript presented in an intelligible fashion and written in standard English?

Reviewer #1: Yes

Reviewer #2: Yes

Reviewer #3: Yes

Reviewer #4: No

6. Review Comments to the Author

Reviewer #1: OK. my past comments are amended and hereby I confirmed the revised version. No more problem.

OK. my past comments are amended and hereby I confirmed the revised version. No more problem.

Reviewer #2: The paper is well organized and eligible to be published. The details are provided very well and it is well written

Reviewer #3: In general, the article has been well described, and structured, and there is an element of novelty. However, some things must be added to make this article more perfect.

Reviewer #4: 1. In the background, regarding the significancy of this study, I suggest the authors add more justification why tourist to tourist interaction is important, and to what extent the interaction is. This is important because in the tourism setting, interaction among tourists are affected by the type of the destination.

2. There are 746 usable questionnaire responses. Suggestions on how the authors could achieve a higher response rate would enhance the study's methodological clarity.

3. Furthermore, elucidate on the specific types of destinations under discussion within this paper.

4. The portion to Destination Sustainability haven't being discussed much in this paper. Please add more discussion about this.

5. It may be beneficial to alter the writing style, as the consistent use of 'we' throughout the paper could be revised to passive constructions for improved clarity and objectivity.

7. PLOS authors have the option to publish the peer review history of their article (what does this mean?). If published, this will include your full peer review and any attached files.

Reviewer #1: **Yes: **Majid Mohammad Shafiee

Reviewer #2: **Yes: **Dariyoush Jamshidi

Reviewer #3: No

Reviewer #4: No

---

## [Author Response · Author response to Decision Letter 1]

21 Jun 2024

Review result

Introduction 

Row 50 (page 3) please outline sustainable destination development with the latest referral support.

Thanks very much for your very constructive suggestions. According to your comment, we have cited some latest article to explain the conception of sustainable tourism. We described “Sustainable development of a destination ” as “a balanced development in the economic, social, cultural, and environmental dimensions of the destination (Crabolu et al., 2023; Streimikiene et al., 2021)”.

Crabolu, G., Font, X., & Miller, G. (2023). The Hidden Power of Sustainable Tourism Indicator Schemes: Have We Been Measuring Their Effectiveness All Wrong? Journal of Travel Research

Streimikiene, D., Svagzdiene, B., Jasinskas, E., & Simanavicius, A. (2021). Sustainable tourism development and competitiveness: The systematic literature review. Sustainable Development, 29, 259–271.

Row 81-83 (page 4) In other words, the fulfillment of basic psychological needs plays a critical mediating role between TTI quality and tourist loyalty (preferably supported by previous research).

We so appreciate for your advice. Based on you suggestion, we have used previous research to explain why the fulfillment of basic psychological needs is a feasible path linking with TTI quality and tourist loyalty in this study. Detailed modification as follow:

“given that the satisfaction of basic psychological needs has been effective transmitting mechanism behind high quality relationships and behavioral outcome (Ahn, 2019),, this study examines the mediating role the satisfaction of basic psychological needs in the influence of TTI quality on tourist loyalty”

Ahn, J. (2019). Role of harmonious and obsessive passions for autonomy, competence, and relatedness support with integrated resort experiences. Current Issues in Tourism, 23(6), 756–769. 

Literature review and hypotheses development

Row 101 (page 5), literature review, and hypotheses development (please add supporting theories as the basis for this research). 

Thanks very much for your very constructive suggestions. We have supplemented the literature review regarding self-determination theory. Detailed content as follow:

“As a comprehensive theory of human motivation, self-determination theory emphasizes the importance of satisfying three basic psychological needs—autonomy, competence, and relatedness—for individual behavior and psychological health (Deci & Ryan, 1985, 2000). Autonomy refers to the feeling that one's actions are self-chosen; competence denotes the feeling of being effective in interactions with the environment; and relatedness indicates the feeling of being cared for and connected to others. SDT posits that a process of individuals internalizing external motivations depends on the degree to which these basic psychological needs are satisfied (Ryan & Deci, 2000). In the field of tourism research, SDT provides a robust framework for understanding and explaining tourists' motivations, behaviors, and satisfaction (e.g., Çiki & Tanriverdi, 2023; Li & Shi, 2022; Thal & Hudson, 2019). Therefore, this study applies self-determination theory to discuss the transmitting mechanism behind the impact of TTI quality on tourist loyalty from the satisfaction of basic psychological needs.”

Deci, E. L., & Ryan, R. M. (1985). Intrinsic motivation and self-determination in human behavior. Springer.

Deci, E. L., & Ryan, R. M. (2000). The “what” and “why” of goal pursuits: Human needs and the self-determination of behavior. Psychological Inquiry, 11(4), 227-268.

Ryan, R. M., & Deci, E. L. (2000). Self-determination theory and the facilitation of intrinsic motivation, social development, and well-being. American Psychologist, 55(1), 68-78.

Çiki, K. D., & Tanriverdi, H. (2023). Examining the relationships among nature-based tourists’ travel motivations, ecologically responsible attitudes and subjective well-being within the scope of self-determination theory. Current Issues in Tourism, 1–6. https://doi.org/10.1080/13683500.2023.2250509

Li, R., & Shi, Z. (2022). How does social support influence tourist-oriented citizenship behavior? A self-determination theory perspective. Frontiers in Psychology, 13. https://doi.org/10.3389/fpsyg.2022.1043520

Thal, K. I., & Hudson, S. (2019). A conceptual model of Wellness destination Characteristics that contribute to Psychological Well-Being. Journal of Hospitality & Tourism Research, 43(1), 41–57. https://doi.org/10.1177/1096348017704498

Row 133-134 (page 7), however, few scholars have investigated the relationship between TTI quality and tourist loyalty (preferably supported by previous research).

We so appreciate for your comments. We have cited previous research to support the viewpoint that few scholars have investigated the relationship between TTI quality and tourist loyalty (Li et al., 2024)

Li, X., Yuan, Y., & Zhang, J. (2024). The influence of tourists’ emotional experiences on destination loyalty from the perspective of community economy. International Journal of Tourism Research. International Journal of Tourism Research, 26(1). https://doi.org/10.1002/jtr.2630

Row 144 (page 7), Thus, TTI quality is likely to positively influence tourist loyalty (must be supported by previous research to strengthen assumptions).

Thanks your very constructive advice. We have quoted previous research to strengthen assumptions that TTI quality is likely to positively influence tourist loyalty (Zhou et al., 2023)

Zhou, G., Liu, Y., Hu, J., & Cao, X. (2023). The effect of tourist-to-tourist interaction on tourists’ behavior: The mediating effects of positive emotions and memorable tourism experiences. Journal of Hospitality and Tourism Management, 55, 161–168. https://doi.org/10.1016/j.jhtm.2023.03.005

Methods 

Row 262 (page 12) It has not revealed the sampling technique used to determine the sample.

We so appreciate for your suggestions. We have supplemented the sampling technique used by this study——convenience sample.

---

## [Decision Letter · Decision Letter 2]

2 Jul 2024

Achieving Destination Sustainability: How Tourist-To-Tourist Interaction Quality Affects Tourist Loyalty?

PONE-D-23-22635R2

Dear Dr. Guo,

We’re pleased to inform you that your manuscript has been judged scientifically suitable for publication and will be formally accepted for publication once it meets all outstanding technical requirements.

Kind regards,

Bo Pu, Ph.D.

Academic Editor

PLOS ONE

Additional Editor Comments (optional):

this manuscript should be published.

Reviewers' comments:

Reviewer's Responses to Questions

**Comments to the Author**

1. If the authors have adequately addressed your comments raised in a previous round of review and you feel that this manuscript is now acceptable for publication, you may indicate that here to bypass the “Comments to the Author” section, enter your conflict of interest statement in the “Confidential to Editor” section, and submit your "Accept" recommendation.

Reviewer #1: All comments have been addressed

Reviewer #3: All comments have been addressed

2. Is the manuscript technically sound, and do the data support the conclusions?

Reviewer #1: Yes

Reviewer #3: Yes

3. Has the statistical analysis been performed appropriately and rigorously? 

Reviewer #1: Yes

Reviewer #3: Yes

4. Have the authors made all data underlying the findings in their manuscript fully available?

Reviewer #1: Yes

Reviewer #3: Yes

5. Is the manuscript presented in an intelligible fashion and written in standard English?

Reviewer #1: Yes

Reviewer #3: Yes

6. Review Comments to the Author

Reviewer #1: OK. well done. all suggested amendments are done, no more changes is needed.

OK. well done. all suggested amendments are done, no more changes is needed.

Reviewer #3: Referring to some of the suggestions for improvement that have been given to the author and the author has provided feedback, then the reviewer can state that: In the introduction, some recent references about the sustainable destination development outline have been added. Then, the role of basic psychological needs plays a critical mediating role between TTI quality and tourist loyalty has been supported by previous research.

In the section on Literature review and hypotheses development. For literature review and hypothesis development, theories have been added to support this research. Then, the relationship between TTI quality and tourist loyalty has been added with relevant latest references to reinforce the assumption. A sampling technique has been added in the method section to determine the number of research samples. So in general this article has met the criteria and is worthy of publication

7. PLOS authors have the option to publish the peer review history of their article (what does this mean?). If published, this will include your full peer review and any attached files.

Reviewer #1: **Yes: **Majid Mohammad Shafiee

Reviewer #3: No

---

## [Editor Report · Acceptance letter]

9 Jul 2024

PONE-D-23-22635R2 

PLOS ONE

Dear Dr. Guo, 

I'm pleased to inform you that your manuscript has been deemed suitable for publication in PLOS ONE. Congratulations! Your manuscript is now being handed over to our production team.

Kind regards, 

on behalf of

Dr. Bo Pu 

Academic Editor

PLOS ONE